

# Carbon stocks and dynamics at different successional stages in an Afromontane tropical forest

Brigitte Nyirambangutse[1,2,*], Etienne Zibera[2], Félicien K.Uwizeye[2], Donat Nsabimana[2], Elias Bizuru[2], Håkan Pleijel[1], Johan Uddling[1], and Göran Wallin[1,*]

[1]Department of Biological and Environmental Sciences, University of Gothenburg, PO Box 461, SE-405 30 Sweden.
[2]Department of Biology, University of Rwanda, University Avenue, PO Box 117, Huye, Rwanda.

*Correspondence to*: Brigitte Nyirambangutse (brigitte.nyirambangutse@bioenv.gu.se) and Göran Wallin
(goran.wallin@bioenv.gu.se)

**Abstract.** As a result of different types of disturbance, forests are a mixture of stands at different stages of ecological succession. Successional stage is likely to influence forest productivity and carbon storage, linking the degree of forest disturbance to the global carbon cycle and climate. Although tropical montane forests are an important part of tropical forest ecosystems (c. 8%, elevation > 1000 m a.s.l.), there are still significant knowledge gaps regarding the carbon dynamics and

15 stocks of these forests, and how these differ between early (ES) and late successional (LS) stages. This study examines the carbon (C) stock, relative growth rate (RGR), and net primary production (NPP) of ES and LS forest stands in an Afromontane tropical rainforest using data from inventories of quantitatively important ecosystem compartments in fifteen 0.5 ha plots in Nyungwe National Park in Rwanda.

The total C stock was 35% larger in LS compared to ES plots due to significantly larger above ground biomass (AGB; 185
and 76 Mg C ha⁻¹ in LS and ES plots, respectively), while the soil and root C stock (down to 45 cm depth in the mineral soil) did not significantly differ between the two successional stages (178 and 204 Mg C ha⁻¹ in LS and ES plots, respectively). The main reasons for the difference in AGB were that ES trees had significantly lower stature and wood density compared to LS trees. However, ES and LS stands had similar total NPP (canopy, wood and roots of all plots ~ 9.4 Mg C ha⁻¹) due to counterbalancing effects of differences in AGB (higher in LS stands) and RGR (higher in ES stands). The AGB in the LS
plots was considerably higher than the average value reported for old-growth tropical montane forest of Southeast Asia and central and South America at similar elevations and temperatures, and of the same magnitude as in tropical lowland forest of different regions.

The results of this study highlight the importance of accounting for disturbance regimes and differences in wood density and allometry of tree species dominating at different successional stages in attempts to quantify the C stock and sink strength of
30 tropical montane forests and how it may differ among continents.



# 1 Introduction

Tropical forests store 40-50% of the carbon in terrestrial biomass (Phillips 1998; Lewis et al., 2009) and account for one third of global terrestrial net primary productivity (Saugier et al., 2001; Malhi et al., 2014;) thereby contributing significantly to the global carbon cycle and climate. In addition to their influence on climate, tropical forests also provide other important ecosystem services such as food, wood products, erosion control, biodiversity protection and water regulation (Costanza et al., 1997; Alamgir et al., 2016). Tropical montane forests (TMF) cover c. 8% (elevation > 1000 m a.s.l.) of the total tropical forest area (Spracklen and Righelato, 2014) and are considered as specifically important for harboring biodiversity and water regulation (Martínez et al., 2009; Scatena et al., 2011). Studies indicate that TMF has been underestimated with respect to its capacity to store (Spracklen and Righelato, 2014) and sequester (Fehse et al., 2002) carbon (C). However, current understanding of the role of TMF in regulating global biogeochemical cycles is hampered by the paucity of field data on productivity and soil carbon, but also biomass, especially from the African continent (Malhi et al., 2013a,b; Spracklen and Righelato, 2014). Our current understanding of carbon storage in tropical forests is to a large extent based on studies of lowland forests in South America and Southeast Asia. Despite of being the world's second largest tropical forest block, African tropical forests have drawn little attention in terms of C cycling research compared to their counterparts in South America and Southeast Asia (Lewis et al., 2009; Malhi et al., 2013a,b).

A recent compilation study indicates that the above ground biomass (AGB) in Central African lowland forest is higher compared to lowland forests in Central and East Amazonia (Lewis et al., 2013), which may be related to differences in climate, soil, biodiverse and/or legacy of disturbance. Ensslin et al. (2015) suggested that AGB may also be higher in African TMF compared to those in South America. However, because of the low number of studies in Africa it is currently difficult to draw any conclusions about the differences in biomass of TMF of the two continents. Even for basic forest attributes such as biomass, stand structure, and species diversity and composition the number of African studies of both lowland and montane forest are sparse (Lewis et al., 2013; Spracklen and Righelato, 2014; Bastin et al., 2015).

There is a growing number of studies conducted in lowland tropical forests, focusing on estimates of net primary productivity (NPP), gross primary productivity (GPP) and carbon allocation (Aragão et al., 2009; Doughty et al., 2014; Malhi et al., 2014). Such studies are still few in TMF, causing large uncertainty regarding the carbon fluxes of these ecosystems (Girardin et al., 2014, Spracklen et al., 2016). In general, TMF are thought to have low productivity compared to lowland tropical forests (Bruijnzeel and Veneklaas, 1998; Girardin et al., 2014; Huasco et al., 2014), but this is not always the case (Fehse et al., 2002). Up to now, no studies of productivity have been reported for African TMF.

Most tropical forests are a mix of disturbed and undisturbed stands, denoted secondary and primary (or old-growth) forests, respectively. Secondary forests are defined as forest regenerating largely through natural processes after significant human and/or natural disturbance (Chokklingham and de Jong, 2001). The detailed distribution of this forest type remains uncertain





since existing information represents substantial variation in terms of categorization with respect to degree of disturbance. However, secondary forests account for at least 40 to 60% of the total tropical forest area and are therefore considered as both ecologically and economically very important (Brown and Lugo, 1990; FAO, 2010) as well as an important part of the global carbon cycle (Birdsey and Pan, 2015; Noormets et al*.,* 2015). Their fraction is expected to increase in several tropical

regions as the human pressure continues to increase over the coming decades (Lewis et al., 2015), e.g. in central Africa (Feintrenie, 2014). In spite of their importance and frequent occurrence, tropical secondary forests have received comparatively little attention as the majority of studies reported this far have focused on old-growth tropical forests (Clark et al., 2001, Malhi et al., 2014). The C uptake and storage of secondary forests is therefore still highly uncertain (Pan et al., 2011). Secondary forests are characterized by high abundance of early successional (ES) tree species, which successively

will be replaced by late successional (LS) tree species dominating in undisturbed old-growth forests. This replacement may take several decades to centuries (Peña-Claros, 2003; Liebsch et al., 2008; Martin et al., 2016) and since the ES species grow faster, but may have a lower stature and wood density ($\rho$) than LS species (Lawton 1984; Poorter et al., 2008; Gustafsson et al., 2016), both the productivity and the forest C stock is likely to be affected during this period. However, although studies of secondary forests indicate that they have high aboveground productivity and carbon sink strength (Sierra et al., 2012),

belowground compartments have rarely been investigated (Berenguer et al., 2014).

Quantification of both biomass and productivity relay on established tree allometric relations. Such relations formulated as equations may be generic (based on multiple tree species harvested in many different sites), site and species specific, or something intermediate (Chave et al., 2005; Chave et al., 2014; Jara et al., 2015). Chave et al., (2005) showed that the most important parameters in estimating biomass in pantropical forests were (in decreasing order of importance) trunk diameter,

$\rho$, tree height (*H*) and forest type. Thus, to substantially improve allometric estimates of forest biomass in African forests more information on these variables is needed for key tree species in different types of forests (Gibbs et al., 2007; Kearsley et al., 2013; Lewis et al., 2013). Height has mostly been omitted in early estimates of tropical forest biomass (Feldpausch et al., 2011), but when included it reduced the standard error from 19.5% to 12.5% in the pantropical biomass estimates (Chave et al., 2005). Moreover, the use of pantropical allometric equations where *H* was not incorporated caused a 52% biomass

overestimation in TMF compared to when *H* was incorporated (Girardin et al., 2010). When *H* is incorporated in the biomass estimations, it is normally calculated from *H vs D* relationships established from measurements on subsamples of trees. Since the *D* vs *H* relationship may vary greatly among forest types and regions, specific information on this trait is critical to accurately estimate forest biomass (Feldpausch et al., 2011; Kearsley et al., 2013). Furthermore, today most studies apply species- or genus-specific $\rho$ data available in a rather comprehensive database (Chave et al*.,* 2009; Zanne et al., 2009).

However, $\rho$ data are still lacking for many important tropical tree species and there may also be considerable variation in this trait within a given species, likely due to variation in environmental conditions among sites (Muller-Landau, 2004). Species-specific on-site information for key species may therefore be valuable, and in studies investigating the influence of successional stage on forest biomass.





With the overall aim to reduce the knowledge gap regarding the carbon balance of African TMF, we quantified carbon stocks and productivity of 15 half-hectare plots with mature trees, but with different disturbance history and different abundance of early and late successional tree species (ES and LS tree species, respectively). We assessed above and below ground carbon stocks, tree recruitment and mortality, and NPP of leaves, wood and roots. We hypothesized that: (1) Tree biomass and total C stock is higher in LS compared to ES stands; (2) trees in ES stands are smaller but have higher relative growth rates compared to trees in LS stands, resulting in similar NPP at both successional stages; (3) it is critical to account for variation in allometric relations and ρ when quantifying and comparing forest biomass at different successional stages, since ES and LS species differ in these traits, (4) carbon stocks are higher in the African TMF studied here compared to TMF in South America.

## 2 Material and Methods

### 2.1 Study area

The study was conducted in Nyungwe tropical montane rainforest located in south-western Rwanda (2°17´-2°50´S, 29°07´-29°26´E), ranging from 1600 to 2950 m a.s.l. Nyungwe forest was gazetted as a National park in 2004 (Gross-Camp et al., 2012). It covers an area of 1013 km$^2$ and is the largest remaining middle elevation montane rainforest in central Africa. It is hosting a large biodiversity, supporting approximately 1105 vascular plant species of which 230 are trees, 280 bird species and is home to 13 species of primates (Plumptre et al., 2007). The forest contains various ecosystems ranging from dense forest and bamboo groves to marshes. Large areas consist of a mixture of primary and secondary forest (Fashing et al., 2007) due to its disturbance history (Plumptre et al., 2002; Masozera and Alavalapati, 2004; Masozera et al., 2006). The secondary forest areas are mainly created from human induced disturbance such as tree cutting, fire, and mining, but natural disturbances such as landslide and fallen trees are also significant. The soils were developed on quartzite schist, mica schist, schist and granite parent material (Cizungu et al., 2014). The mineral top soil consists of clay, sand and silt ranging from 2 to 71%, 9 to 61%, 5 to 61% with averages of 34, 43 and 23%, respectively (Gharahi Ghehi et al., 2014). At a meteorological station located at Uwinka (2° 28′ 43″ S, 29° 12′ 00″ E, 2465 m a.s.l. elevation, Nsabimana, 2009), the average day and night air temperatures were 15.7 °C and 13.5 °C, respectively, the relative humidity was 81%, and annual rainfall was 1867 mm during 2007 - 2015. The difference between the warmest and coldest month was 1.1 °C. There is a two month dry season normally occurring from mid-June to mid-August.

### 2.2 Plots

In late 2011 and early 2012, 15 permanent plots with a planimetric area of 0.5 ha (100 x 50 m) were established. The plots were arranged along a 32 km long west-east transect at an elevation of c. 1950 to 2500 m a.s.l. Forest stands ranging from a dominance of early successional (ES) to a dominance of late successional (LS) species were included, but areas with recent and extensive disturbance were excluded. The most abundant ES and LS tree species were *Macaranga kilimandscharica* Pax



and *Syzygium guineense* (Engl.) Mildbr., respectively. Each plot was subdivided into eight subplots with a size of 25 m x 25 m. All individual woody plants with a breast height diameter ($D$) ≥ 5 cm were mapped and identified to species level when possible. The total number of identified tree species was 83. A subset of species consisting of those that were among the four most abundant species with respect to basal area in any of the 15 plots was selected for more detailed studies to facilitate the

estimation of C stock and productivity. This subset comprised in total 22 species representing 90% of the basal area and 79% of all individual stems across all 15 plots. Plot positions and climate are given in Table 1 and stand characteristics are presented in Table 2.

## 2.3 Meteorological data

Data on air temperature, air humidity, solar radiation, humidity and precipitation were collected every 30 min from four

meteorology stations installed along the transect of plots (Table 1). One major station was established at the Uwinka research site in February 2007 (Nsabimana, 2009) and three minor additional stations were established in June 2013. The Uwinka station was installed in a 15 m tower at a hill top to reach above the canopy while the others were installed at open areas at 3 m height (1.5-2 m above ground vegetation). The minor stations were equipped with sensors for measurements of temperature, relative humidity, solar radiation and precipitation (VP-3, PYR and ECRN-100, respectively, from Decagon

Device, Inc (Pullman, WA, USA) connected to a data logger (Em50G). At one of these stations soil temperature and moisture was also measured by a combined sensor (5TM). Air temperature and humidity was also measured at the centre of each plot (under the canopy) by using mini-loggers (Model TinyTag Plus 2, Gemini data loggers Ltd, United Kingdom) placed inside self-ventilating radiation shields at approximately 3 m above the ground.

## 2.4 Stem mass and NPP

The diameter at breast height ($D$) of all trees with a $D \geq 5$ cm was determined using diameter tape in two censuses in: (1) October – July 2011/12 and (2) October – June 2014/15, respectively, i.e. on average 3 years in between. For trees with major irregularities (e.g. buttresses) at breast height, the point of measurements (POM) was moved upwards the stem (max 6.5 m above ground). The $D$ of these trees was estimated using the taper function by Metcalf et al. (2009):

$$D = \frac{D_h}{e^{(-\alpha(h-1.30))}} \qquad (1)$$

where α = 0.31, as determined for 31 African tropical species by Ngomanda et al. (2012) and $D_h$ is the stem diameter at height, $h$. The height ($H$) of 930 trees, representing the full $D$ range of the most abundant species in the subset defined above, was measured using a clinometer (Vertex IV, Haglöfs Sweden AB, Långsele, Sweden). To estimate the tree $H$ of all individuals of these species, $H$ $vs$ $D$ relationships specific for each species were established by fitting data to the following function by Lewis et al. (2009):



$$H = a(1 - e^{(b(D^c))}) \qquad (2)$$

where a, b and c are fitting parameters. For all other species, we used generic parameters obtained by fitting Eq. (2) to the data representing all measured trees. Wood density (ρ) of the most abundant tree species was estimated by taking wood cores at breast height by using an increment borer (Haglöf Sweden AB, Långsele, Sweden). The diameter (5.15 mm) and length

(below bark to center of stem) of the fresh cores and the mass of the oven dried (70 °C) cores were used to calculate the ρ (g cm$^{-3}$). When presented as plot mean, BA-weighted ρ was used ($\rho_{BA}$). The stem mass (including branches) was estimated using the equation of Chave et al. (2014):

$$M_{\text{Stem}} = 0.0673 \, (\rho D_{\text{BH}}{}^2 H)^{0.976} \qquad (3)$$

where $M_{\text{Stem}}$ is the biomass of individual stems in kg m$^{-2}$, and $H$ is tree height (m). The coarse root mass ($M_{CR}$) was estimated

based on $M_{\text{Stem}}$ using median root to shoot ratio from Cairns et al. (1997) as follows:

$$M_{CR} = 0.21 \, M_{Stem} \qquad (4)$$

To convert biomass into carbon (C) mass, we assumed a carbon concentration of 47.4% in line with Martin and Thomas (2011). The net primary production of stems (NPP$_{\text{Stem}}$, Mg ha$^{-1}$ yr$^{-1}$) was calculated according to the following equation:

$$\text{NPP}_{\text{Stem}} = \left( \frac{\Sigma M_{\text{Stem2}} - \Sigma M_{Stem1}}{\Delta t} \right) A^{-1} \qquad (5)$$

where $\Sigma M_{\text{Stem1}}$ and $\Sigma M_{\text{Stem2}}$ is the sum of $M_{\text{Stem}}$ at census 1 and 2 in a plot of area A and, $\Delta t$ is the time in between the census (2.5 to 3.7 years). Relative growth rate (RGR$_{\text{Stem,}}$ % yr$^{-1}$) is calculated as follows:

$$\text{RGR}_{\text{Stem}} = \left( \frac{lnM_{\text{Stem2}} - lnM_{Stem1}}{\Delta t} \right) \times 100 \qquad (6)$$

for each individual stem, thereafter RGR is averaged for a certain area.

To monitor the stem growth of the two most abundant species *M. kilimandscharica* and *S. guineense* in detail, dendrometer

bands (Jädraås skog och mark, Jädraås, Sweden) were installed at breast height or higher (see *D* measurements above) on 125 trees of each species. A randomized block design was applied where *D*-classes (10 cm intervals) and plots were used as blocks. The increments were observed approximately every four months with calipers. The averages of these readings were used to calculate the annual increment of $M_{\text{Stem}}$ and RGR$_{\text{Stem}}$.





### 2.5 Canopy NPP

Litter from 90 traps distributed over all plots where collected twice per month from January 2013 to December 2014. In each plot, six of the subplots were randomly assigned one trap that was randomly placed at one of 16 grid points within each subplot using a 5 x 5 m grid. The litter traps consisted of nylon mesh bags suspended from a circular wire frame of

aluminum (0.3 m$^2$; Jädraås skog och mark, Jädraås, Sweden) and mounted horizontally on wooden poles ca 0.8 m above ground level. The litter from each trap was collected separately, placed in paper bags, and sent to the lab where it was oven-dried at 70$^{\circ}$C to constant mass. After drying, each sample was sorted into five fractions: leaves, reproductive organs (fruits, flowers, seeds), twigs, epiphytes, and unidentified fine debris, and weighed. The annual sum of all five fractions was used to calculate the canopy NPP.

### 2.6 Fine root, litter and soil organic mass

Litter and soil were sampled from the centre of each subplot quadrant (480 sample points), where the litter and organic (O) soil was separately excavated from a 0.5 x 0.5 m horizontal ground area. Below the O-horizon, three consecutive cores (8 cm diameter and 15 cm depth each) of mineral (M) soil down to a depth of 45 cm were sampled using a root auger (Ejkelkamp soil & water, Giesbeek, The Netherlands), thereafter the four M-samples within each subplot and depth was mixed. A

subsample of 20% based on the fresh mass was taken from each O- and mixed M – sample, thereafter roots were extracted from each subsample. All samples were then brought to the lab for drying to constant mass in oven set to 70 °C. The litter and soil samples were milled in a ball mill (Model: MM 200, Retsch, Germany) and C concentrations were determined by dry combustion using an elemental analyzer (Model: EA 1108 CHNS-O, Fisons Instruments, Italy).

### 2.7 Fine root NPP

The fine root production was measured using in-growth cores with root free soil surrounded by mesh-containers (40 cm deep, 8 cm diameter and c. 2 mesh, i.e. 12 mm grid). The in-growth cores were installed in the soil by drilling a vertical hole of 8 cm in diameter to the depth of 40 cm in the middle of each subplot, using a root auger (see above). The soil from the drilling was separated into the O- and M-horizons, and after removing the roots it was used to fill the mesh container installed into the drilling pit, maintaining the soil horizons. The in-growth cores were installed in September and December

2013, and fine-roots were allowed to grow into the cores during periods of 3-6 months before harvested in March 2014, July-August 2014, December-January 2014/15 and July 2015. To avoid underestimation of root mass because a proportion of the roots inevitably remain uncollected (Sierra et al., 2003), this study followed the method by Metcalfe et al. (2007) which controls for systematic underestimation of fine roots. This was conducted by extracting the roots from the soil in the in-growth cores during four 8 min intervals (32 min), considering O-and M-horizons separately, and then fitting the cumulative

increase of collected root mass over time to the following equation to predict root mass as if the extraction was continued during 120 min:





$$M_{fr,t} = a \log(t) + b \tag{7}$$

where $M_{fr,t}$ is the fine root mass extracted at time $t$; a and b are fitting parameters. The roots were brought to the lab and cleaned from soil by rinsing and sedimentation processes in tap water and thereafter dried in 70 °C until constant mass. The annual sum of production was used as fine root NPP.

**2.8 Understory**

The understory defined as all aboveground parts of plants with a $D < 5$ cm (woody, herbaceous and grass species) were sampled from one square meter plot (1 x 1 m) randomly placed at one of 16 grid points within each subplot using a 5 x 5 m grid. All plants within the one square meter plots were harvested at ground level and thereafter dried in 70 °C until constant mass, from which the dry mass per area of understory ($M_{Ustory}$) was calculated. Based on $M_{Ustory}$ and $M_{Stem}$, an understory
index (UI) was developed to classify the fraction of understory biomass:

$$UI = \frac{M_{Ustory}}{(M_{Ustory} + M_{Stem})} \tag{8}$$

**2.9 Successional index**

A successional index (SI), ranging from 0 to a maximum of 1, was developed to classify the successional stage of the plots from the fractions of ES and LS trees within the plots:

$$SI_x = \frac{LS_x}{T_x} \times \left(1 - \frac{ES_x}{T_x}\right) \tag{9}$$

where T is the plot total. The subscript x denotes if it is based on basal area (BA) or number of tree individuals (#). In this study we based the index on the 10 most abundant species representing 77% of the basal area and 59% of the individuals of all plots (see Table 2). The classification of which successional group the species belongs to was mainly based on Tesfaye et al. (2002), Fischer and Killman (2008), Bloesch et al. (2009), Kindt et al. (2014) and Rutten et al. (2015a) and we found that
three belong to ES and seven to LS (Table 2). Based on this index, two groups of five plots each were defined, one with the lowest (< 0.1) and one with the highest $SI_x$ (> 0.5), denoted ES and LS plots, respectively. Similar ranking and grouping resulted if the index was based on number of stems instead (Table S1). However, the $SI_x$ values using number of stems were generally lower since many of the frequently occurring but small trees were not classified for successional groups. The five plots having a $SI_x$ between the ES and LS groups were classified as intermediate successional plots (MS).





## 2.10 Recruitment and mortality

The recruitment rate ($\lambda$, % yr$^{-1}$) and mortality rate ($\mu$, % yr$^{-1}$) was determined from the number of stems (> 5 cm $D$) in census 1 ($n_0$), in census 2 ($n_t$) and the number of stems that died ($D_t$) over the time between the two censuses ($t$). The rates were calculated according to Sheil and May (1996), including a correction factor suggested by Lewis et al. (2004), as follows:

$$\lambda = \left[ \frac{\ln n_0 - \ln(n_0 - D_t)}{t} \times 100 \right] \times t^{0.08} \tag{10}$$

$$\mu = \left[ \frac{\ln n_t - \ln(n_0 - D_t)}{t} \times 100 \right] \times t^{0.08} \tag{11}$$

## 2.11 Statistics

The significance of the relationship between biomass and production parameters *versus* the successional indices, understory index, number of big trees and basal area were determined using the regression analysis tool in SigmaPlot 12.5 (Systat Software Inc., San Jose, CA, USA). Differences in forest structure, biomass, C stock and productivity between early and late successional plots and species were analysed using two-tailed independent-samples t-test using SPPS software (IBM SPSS Statistics for Windows, Version 22.0. Armonk, NY: IBM Corp.). Data that violated the assumption of normality or when outliers were present was log-transformed before the statistical analysis.

## 3 Results

### 3.1 Climate

The climate was monitored along the transect of plots between July 2013 and June 2015 and was characterized by very small seasonal variations in monthly mean air temperature (c. 1°C, Fig. 1a), while the seasonal variation in precipitation exhibited a short dry period in July and less than average from May to August (Fig. 1b). The annual mean air temperature under the canopies at plots 1 to 12 varied between 13.7 ± 0.002 to 14.5 ± 0.07 °C and had an annual precipitation of 1657 ± 163 to 1860 ± 116 mm (Table 1). Plots 13-15 located at c. 500 m lower elevation compared to the others had a mean air temperature of 15.5 ± 0.06 °C and an annual precipitation of 3016 ± 63 mm during the same period. The lowest monthly mean daily minimum temperature varied between 9.9 to 11.7 °C and the highest monthly mean daily maximum varied between 17.7 to 21.2 °C among plots. The average temperatures measured at climate stations in open areas nearby plots were on average 0.3 °C higher than under the canopy. The soil water content at the end of the dry periods varied between 0.05 - 0.1 m$^3$ m$^{-3}$ (at 10 cm depth) while it normally varied between 0.25 - 0.4 m$^3$ m$^{-3}$ outside the dry period. The two years of climate data presented for the transect was similar to the 9 years average from our long-term monitoring station.





### 3.2 Successional stage

The successional index used (SI; Eq. (9)) to characterize the differences in successional stages was markedly different between ES and LS plots using both BA (0.03 and 0.65, respectively) and number of stems (#) per area (0.03 and 0.33, respectively; Table 2) as an index basis. The highest SI in any plot was 0.89 and the reason that the maximum SI (1) was not reached in any of the plots is partly due to the occurrence of some ES species in all plots, but mainly because not all tree species were classified into SI groups. The SI is conservative towards the ES forest type and is sensitive to fact that a high degree of non-classified species gives lower values. The classification into ES and LS groups was used to test for differences in forest structure, carbon stock and productivity between stands of different successional stages (Table 2) and in the following, the values reported for the two groups of plots will be separated by an oblique (/) in the order ES/LS.

### 3.3 Forest structure

Many of the common forest structure parameters determined as means for all trees with a $D > 5$ cm (e.g. stem density, tree cross sectional area at breast height, BA, $H$) were generally higher in the LS plots compared to ES plots but the difference was not significant ($P \geq 0.12$; Table 2). However, the $H$ of the 100 highest trees per ha (22.2/26.9 m; $P = 0.04$), indicating the canopy height, and the BA weighted $\rho$ (ES: 0.48, LS: 0.62 g cm$^{-3}$), $P < 0.001$) were significantly higher in LS compared to ES plots. The total number of woody species with a $D > 5$ cm was 83 (Table S2), with an average of 23 per plot and no significant difference between ES and LS plots (17/29, $P = 0.093$). However, the abundance based on BA of several species was substantially different, but significantly so only in a few cases due to large variation in species composition between plots also within successional groups (Table 2).

The distribution of stem number across $D$ classes in all forest types described an exponential decay function with increasing $D$ and thus decreased linearly when a logarithmic scale for stem numbers was applied (Fig. 2a). Notably, ES plots were lacking stems in $D$ classes > 90 cm. Generally, trees with a $D$ of 5 to 10 cm had a small contribution to the total biomass (< 2.5%; Fig. 2b). The distribution of biomass across $D$ classes differed between ES and LS plots, with a major part of stand biomass in relatively small size trees in ES plots (68% in trees < 50 cm $D$) and large size trees in MS and LS plots (> 50% in trees > 50 cm $D$; Fig. 2b). Thus the tree demography varied between ES and LS stands, reflecting a difference in disturbance history.

### 3.4 Biomass and Carbon stock

The above ground C pool of trees was estimated from $D$ measurements of all stems, $H$ measurements of 930 trees of different $D$-classes from the most abundant species, and species specific $\rho$ obtained from measurements (species representing 91.2 % of the BA) or databases (Table S2 and S3). The $H$ measurements were used to determine both a generic (but site specific) and species specific parameterizations of Eq. (2) (Fig. 3). Both the measured data and the generic




parameterization clearly show that the *H* to *D* ratio was lower in this montane forest compared to a generic equation for the central African lowland (Fig. 3a). The output from the species specific parameterization show that ES compared to LS species had significantly lower *Hi)* at *D* of 40 (-10%, *P* = 0.033) and 80 cm (-21%, *P* = 0.020) (Fig. 3b and c). Furthermore, the average ρ of ES compared to LS species was 19% lower, but the difference was only marginally significant (P = 0.057)

5    for the ten most abundant species. However, the BA weighted ρ of LS compared to ES plots was 29% higher (P < 0.001; Table 3). The use of species specific rather than generic *H vs D* relationships and ρ values did, as expected, not change the estimated average stem biomass across all plots (274 Mg ha$^{-1}$) but had a large effect on the estimated difference in stem biomass between ES and LS plots (Table S1). Using the species specific parameters resulted in 146% higher stem biomass in LS compared ES plots (156/387 Mg ha$^{-1}$, *P* = 0.022) while the generic parameters suggested only a 79% difference (191/342

10    Mg ha$^{-1}$, *P* = 0.083).

The total above and below ground carbon pools (down to a depth of 45 cm in the mineral soil, excluding standing and fallen dead wood) averaged to 353 ± 138 Mg C ha$^{-1}$ across all plots, with a non-significant difference between ES and LS pools (299/402 Mg C ha$^{-1}$, *P* = 0.11; Table 4). However, the relationships of C stock$_{Stem}$ with SI$_{BA}$ (R$^2$ = 0.67, Fig. 5 a) and SI$_\#$ (R$^2$ = 0.42, Fig. S1) were both highly significant (P < 0.001). The woody carbon pools were significantly different between ES

and LS plots (C stock$_{Stem}$ 74/183 Mg C ha$^{-1}$, *P* = 0.023; C stock$_{CoarseRoots}$ 16/35 Mg C ha$^{-1}$, *P* = 0.031), as were also AGB (including understory), BGB (including fine roots) and the ratio of AGB to total C (25/44 %, *P* = 0.019; Table 4) and total biomass (AGB + BGB) to total stand C (32/54 %, *P* = 0.020). The C stock$_{Stem}$ was negatively and positively related to understory index (Eq. (8)) and the number (#) of big trees (Fig. 6a, b), respectively, although none of these parameters significantly differed between plots belonging to the two successional groups.

The total C stock of litter, organic soil and mineral soil were similar in the plots of the two successional groups (204/178 Mg C ha$^{-1}$, *P* = 0.27). The depth of the soil organic layer was on average 11 cm (range: 4.5 – 17.4 cm) with no significant difference between ES and LS plots (P = 0.57). Despite the relatively thick organic layer and with high C concentration (26-50%), the C Stock$_{OrganicSoil}$ was relatively low (26/36 Mg C ha$^{-1}$, *P* = 0.34) because of a very low bulk density (0.08 g cm$^{-3}$) as the organic soil horizon mainly consisted of a soft matrix of fine roots and decaying litter.

**3.5 NPP, RGR, mortality and recruitment**

The sum of NPP of different forest compartments (NPP$_{Tot}$) was on average 9.41 ± 1.50 Mg C ha$^{-1}$ yr$^{-1}$ across all plots (Table 5). The variation between plots ranged from 6.7 to 12.1 Mg C ha$^{-1}$ yr$^{-1}$, but no difference between ES and LS plots was observed (9.3/9.2 Mg C ha$^{-1}$ yr$^{-1}$, *P* = 0.93). The ratios of NPP$_{wood}$, NPP$_{FineRoots}$ and NPP$_{Canopy}$ to NPP$_{Total}$ were on average 0.39, 0.21 and 0.40, respectively, and did not significant differ between ES and LS plots (P > 0.32). NPP$_{Stem}$ was related

neither to SI$_{BA}$ (R$^2$ < 0.01; Fig. 5) nor to SI$_\#$ (R$^2$ = 0.04; Fig. S1). However, RGR$_{Stem}$ was negatively related to both SI$_{BA}$ (R$^2$ = 0.27, *P* = 0.048; Fig. 5) and SI$_\#$ (R$^2$ = 0.52, *P* = 0.003; Fig. S1), and RGR$_{Stem}$ was 52% higher in ES compared to LS plots



(8.4/4.1%, yr$^{-1}$, $P$ = 0.017; Table 2). The lack of difference in NPP$_{stem}$ between ES and LS stands is probably the net result of counteracting effects of differences in stem biomass (higher in LS stands) and RGR$_{stem}$ (higher in ES stands).

To explore how production and RGR varied with tree sizes, the growth of the two most abundant ES and LS species (*M. kilimanscharica* and *S. guineense,* respectively) were analysed in detail using dendrometer band readings over three years
(Fig. 4). The stem volume increment and RGR was significantly higher ($P$ = 0.012 and 0.027, respectively) in *M. kilimanscharica* than in *S. guineense* within the *D* range of 10-70 cm. However, the stem mass production did not differ between the two species ($P$ = 0.62, Fig. 4) since *M. kilimanscharica* had lower ρ than *S. guineense* (0.44 and 0.63, respectively). Species specific tree growth within given *D* classes did not significantly vary among plots or along the plot transect (data not shown), indicating that the species specific responses was not constrained by changes in the plot
environment.

The annual recruitment of new stems (6.3/1.4%, $P$ = 0.007) was significantly higher in ES compared to LS plots, while the annual mortality (1.1/1.4%, $P$ = 0.26) was similar in both successional groups (Table 3). However, the most dominant ES species *M. kilimandscharica* had significantly higher annual mortality compared to the most dominant LS species *S. guineense* (2.1/0.63%, P = 0.035) at plots (n = 8) where they co-occurred (≥ 10 stems of each). However, recruitment rate
(3.0/1.4%, $P$ = 0.17) did not differ significantly differ between these species when analysing plots where they co-occurred.

**4 Discussion**

We report here the first comprehensive estimates of the productivity, biomass and carbon stocks of African TMF. We found generally high C stocks and productivity, with higher AGB in later compared to early successional forests but similar productivity across different successional stages. Our results further demonstrated that accurate quantification of the carbon
stocks and dynamics of the forests in the present study required local information on tree allometry, wood density and species composition. This highlights the need to account for such variation in traits when estimating carbon stocks of tropical rainforests at different successional stages and in different regions, as well as when implementing REDD+ projects.

**4.1 Biomass and carbon stocks in relation to successional status**

While ES forest stands with closed canopy and mature trees had significantly lower AGB and BGB (59%, $P$ = 0.023 and
52%, $P$ = 0.025, respectively) than LS stands, there were no significant differences observed for the soil C stock (Table 4). As a consequence, the plant (AGB + BGB) fractions of the total C stock was significantly lower in ES compared to LS plots (32% in ES, 54% in LS; $P$ = 0.020). This finding is in line with other studies reporting relatively unaffected soil C stocks in moderately disturbed and secondary tropical forest (Martin et al., 2013), although disturbances may have significant effects on the soil C stock in steep slopes and thus more severely affect TMF. As shown by the significant relationship between the



stem biomass and the successional index (SI$_x$) based on BA or stem density (Fig. 5a & Fig. S1a) the difference in AGB is connected to the species composition of both ES and LS forest stands. The larger AGB in LS compared to ES forest stands was due to LS tree species having significantly higher $\rho$ and $H$ to $D$ ratio at a given stem diameter (in the larger diameter classes; Fig. 3; Table 3; see 4.3), as well as LS stands having a larger fraction of large trees than ES stands (Fig. 2; Fig. 6b). Basal area did not significantly differ. Our results agree with earlier studies showing that information on the abundance of large trees is an important indicator and determinant of whole forest stand AGB (Slik et al., 2013; Bastin et al., 2015). In summary, our data support hypothesis #1 "Tree biomass and total C stock is higher in LS compared to ES stands".

## 4.2 Productivity and carbon dynamics

Our results are in line with the general observation that ES species grow faster than LS species (Poorter et al., 2008; Gustafsson et al., 2016) since we observed a 53% higher RGR ($P = 0.012$) of the ES dominat species (*M. kilimandscharica*) compared with the LS dominant species (*S. guineense*) over a wide range of $D$ classes. Furthermore, this was expressed also at the plot level as the average RGR in ES plots was twice as high as in LS plots ($P = 0.017$). In spite of the higher average RGR in ES compared to LS species/plots, we observed no significant differences of NPP of any compartment (wood, canopy, fine roots, Table 5) between ES and LS plots. There are likely two reasons for this apparent discrepancy, firstly counterbalancing effects of differences of the AGB (higher in LS) and RGR (higher in ES) on NPP, and secondly the larger amounts of big trees (having lower RGR, Fig. 4c) in the LS compared to the ES plots (Fig. 2a, b). Thus, we find support for hypothesis #2: "trees in ES stands are smaller but have higher relative growth rates compared to trees in LS stands, resulting in similar NPP at both successional stages".

It is generally thought that the productivity of high-elevation rainforests is low, likely due to low temperature and radiation (i.e. increased cloudiness; Bruijnzeel and Veneklaas, 1998). Indeed, productivity declines with increasing elevation have been reported from Borneo and the Andes (Kitayama and Aiba, 2002; Girardin et al., 2010), as well as in a compilation of NPP data from different continents and elevations (Malhi et al., 2011). However, the TMF of the present study has high productivity. Both the above ground and total NPP of this study were only slightly lower (-7% and -17%, respectively) than the average recorded NPP for lowland (< 1000 m a.s.l.) tropical forests (Malhi et al., 2011). When compared with studies at elevations and mean annual temperatures similar to in our study from Malhi et al., (2011), the Nyungwe TMF had among the highest NPP values estimated for TMF. Also Fehse et al. (2002) reported high above ground biomass production of secondary tropical TMF, although not expressed as NPP and therefore difficult to compare. However, our values of stem and canopy NPP were much higher compared to a study by Girardin et al., (2010) in the Andes, while values of fine root NPP did not differ.



### 4.3 Stem allometry and wood density

Parameterisation of equations to establish *H vs D* relationships developed from data obtained from outside the study area causes large uncertainties in C stock estimation of tropical forests (Feldpausch et al., 2011; Kearsley et al., 2013). Indeed, we found significant differences in C stock estimates when using species specific parameters compared to generic parameters of

ρ and *H* to *D* relationship (Table 3; Fig. 3). In particular, the use of species specific parameters greatly affected the comparison of C stocks of LS and ES stands. The stem C stock was 147% higher in LS compared to ES forest stands using local and species-specific parameters, while the difference was reduced to 79% if generic parameters were used (Table 3). With species specific parameters the difference was significant ($P = 0.021$), while it was not if generic parameters were used ($P = 0.083$). The main reasons for this discrepancy were that LS tree species had considerably higher ρ and *H vs D*

relationship (in larger size classes) compared to ES tree species (Table 3; Fig. 3). These results demonstrated that species composition together with differences in ρ and allometry between LS and ES tree species are important drivers for the difference in C stocks between primary and secondary forests.

Our results further showed that the application of lowland *H vs D* relationships (Fig. 3a; lowland parametrization of Eq. (2); Lewis et al., 2009) for our TMF caused considerable overestimation of C stocks (Fig. 3a) as it provided strongly

overestimated *H* for our study area. Thus, our results demonstrate the importance of using locally derived or validated *H vs D* relationships in TMF C stock estimation, as previously also shown in mixed secondary forests of Indonesia (Ketterings 2001).

We found that large LS trees were significantly taller than large ES trees at a given tree diameter, while it was opposite for small sized trees (Fig. 3b, 3c). ES forest stands are characterized by pioneers, with low stature and light wood (Swaine &

Baltimore, 1988; Muller-Landau, 2004). For example, the dominant ES species *M. kilimandscharica* (59.4 ± 14.8 of BA, Table 2) rarely grows taller than 25 m (Fig. 3c). Such differences between ES and LS species have been observed earlier (e.g. King, 1996) and may reflect a trade-off between fast height extension and mechanical stability during forest succession (Lawton, 1984). ES species have faster height growth in the beginning of successional progression but will eventually be surpassed by LS trees with higher ρ and stability.

High relative growth rates and low ρ of ES tree species agrees with the plant economics spectrum of fast-growing species (Woodall et al, 2015; Reich, 2014). The trade-off between ρ and growth rates has been observed also in earlier tropical studies (Poorter et al., 2008, Keeling et al., 2008; Poorter et al., 2010,). Chave et al. (2009) further argued that mortality rates in tropical forests are associated with low density wood, but this was not confirmed by our results where we did not find any significant difference in mortality rates between ES and LS plots (Table 3). However, *M. kilimandcharica*, the most

abundant ES species, had significantly higher mortality rate than *S. guineense*, the most abundant LS species.

Overall the observations in this study support hypothesis #3: "it is critical to account for variation in allometric relations and ρ when quantifying and comparing forest biomass at different successional stages, since ES and LS species differ in these traits". It is therefore likely that a full recovery of the carbon stock not will be achieved before the first generation of ES trees have been replaced by LS trees.

**4.4 Biomass and carbon stocks in relation to studies of old-growth TMF**

To put our results into perspective, we compiled data from studies of AGB (trees with $D > 10$ cm) of old-growth tropical lowland forest (< 1000 m a.s.l.) and TMF (elevations: 1600 to 2800 m a.s.l. and mean annual temperature: 11 to 18 °C, c. ± 300 m and ± 2.5 °C of our lowest and highest elevation and temperature, respectively; Table 6). The late successional (LS) stands of Nyungwe TMF were shown to have somewhat higher AGB than the average old-growth lowland tropical forests of

Central/East Amazonia (+11%) and a bit lower than lowland forests of Central Africa (-11%) and Borneo (-15%; Table 6). Average stem density, basal area and ρ in our LS stands were similar to those observed in lowland tropical forests. These findings support the suggestion by Spracklen and Righelato (2014) that TMF biomass may store more carbon than earlier expected.

Compared to the average AGB of the TMF in Southeast Asia and Central and South American, our LS plots had

substantially higher AGB (Table 6). Only two plots from the studies in Central and South America (Delaney et al., 1997; Alvarez-Alteaga et al., 2013) had similar AGB as our average LS AGB (395 and 408 Mg ha$^{-1}$). However, the AGB in these studies may be overestimated as none of them used $H$ as a parameter in the AGB calculation, an omission which may overestimate the biomass in TMF (Girardin et al., 2010). When we applied the allometric equations from Delaney et al. (1997) to our data we obtained a 20% higher AGB than by using the equation from Chave et al. (2014) including $H$ as a

parameter (see further discussion in 4.3). The only study of undisturbed TMF at the selected altitudes and temperatures from the African continent that we found (Ensslin et al., 2015; Rutten et al., 2015b) reported lower AGB (275 Mg ha$^{-1}$) than in our LS plots but higher than average AGB in the American and Asian TMF studies (Table 6). Although AGB data from African TMF are still scarce, the general pattern of higher AGB in Africa compared to America observed in tropical lowland rainforests (Banin et al., 2012; Feldpausch et al., 2012; Lewis et al., 2013) thus may to hold also for TMF, as also suggested

by Ensslin et al. (2015).

TMF soils are considered to be richer in soil C content than lowland tropical forest (Roman et al., 2011) which has on average 46% of the total ecosystem C stock in the soil down to 1 m (Malhi et al., 2015). Taking into account that an average tropical forest soil has around 25% of the C content between our sampling depth (average 0.56 m) and 1 m (Jobbágy and Jackson, 2000), the soil C fraction in the LS plots was 51%, which was similar to an study in an old-growth TMF in

Venezuela observing 52% (Delaney et al., 1977). Only a few TMF studies have reported the C stock of both below and above ground compartments, which makes it difficult to assess the portioning of C in these ecosystems. In addition, the




below ground C stock of different studies are probably less reliable to compare than above ground C stocks. This is because of inconsistent sample methodology (Roman et al., 2011), such as sampling depth and intensity, differs between studies, combined with the likeliness of large local variation in below ground C stocks, especially in mountain areas where the vertical soil profile is likely to be very variable due to variation in e.g. sloping degree.

Based on available data, our total below ground C stock is similar to the levels reported for studies in South American TMF, when adjusted for differences in sampling depth (Delaney et al., 1977; Grimm and Fassbender, 1977; Girardin et al., 2010; Moser et al., 2011). But, the partitioning of below ground C in the present study differs greatly from the studies by Girardin et al. (2010) and Moser et al. (2011) which were the only studies providing detailed compartmental C stock information. In our study, the C stock of the mineral soil down to 30 cm depth was more than three times as large as in the Andean TMF,

while the C stock of the soil organic layer plus fine roots was only half. The main reasons for this difference in soil C partitioning were that the mineral soil in the present study had very high C concentrations (4-7% in top 30 cm) while the organic soil layer was more shallow (11 cm) compared to the Andean studies (31 cm).

Overall, the observation of high above and below ground C stocks in our LS plots, as well as in Ensslin et al. (2015) support hypothesis: #4 "carbon stocks are higher in the African TMF studied here compared to TMF in South America". However,

ecosystem C stock data from African TMF studies are still scarce and there is a risk for biased results among TMF studies as the average plot size in TMF studies is small (0.22 ha in our compilation).

## 5 Conclusion

The results of this study demonstrate that tropical montane forests in central Africa contain large amounts of C. Undisturbed forests (i.e. LS stands) have higher AGB than their South American montane counterparts; a finding consistent with

comparisons of lowland neo- and paleotropical forests (Malhi et al., 2013b). The study also showed that the AGB was greater in LS than ES stands, with both species composition and stem properties (density and allometry) explaining this difference. Both ρ and the $H$ to $D$ ratio of large trees were greater in LS than in ES tree species, highlighting the importance of accounting for these differences when quantifying the C stock of TMF at different successional stages. Although LS and ES forests differed in AGB, they had similar NPP. This was the net result of counterbalancing effects of differences in

biomass (higher in LS) and relative growth rates (higher in ES). While more work is needed to understand and estimate the C stocks and dynamics of African tropical montane forests, this study provides an important contribution towards that goal and highlights the importance of accounting for disturbance regimes and forest successional stage.




## Author Contributions

BN and GW planed and designed the experiment with important contributions from DN, EB and HP. EZ, BN, FU and GW conducted the field measurements and data compilations. All authors contributed to the analysis and interpretation of the results. BN and GW wrote the manuscript with HP and JU providing important editorial advice. All authors read and approved the final manuscript.

## Acknowledgment

Many thanks are due to Innocent Rusizana, Pierre Niyontegereje, Jean Baptiste Gakima, Ferdinand Ngayabahiga and Isaacar Ndayisabye for great assistance of the field work and to Mats Räntfors for lab assistance. The study was made possible by financial support from University of Rwanda - Sweden program for Research, Higher Education and Institutional Advancement, financed by Swedish International Development Cooperation Agency (Sida). We are also grateful to Rwanda Development Board (RDB) for authorization of conducting research and collection of specimens in Nyungwe National Park.

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



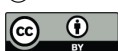

**Table 1.** Temperature (*T*) and precipitation at five locations along a 32 km transect with 15 plots in Nyungwe tropical montane forest during July 2013 to June 2015. Temperature at 3 m above ground was measured both below the canopy in each plot and at four meteorology stations in open areas. SD of plot and meteorology station data represents variation among plots and months, respectively. Plot numbers at each location are given in brackets.

| | Locations (plots): | | | | | | | | | |
| | I (1 - 3) | | II (4 - 6) | | III (7-9) | | IV (10-12) | | V (13-15) | |
| Properties | Mean | SD | Mean | SD | Mean | SD | Mean | SD | Mean | SD |
| Latitude | 2°31'54"S | | 2°31'25"S | | 2°28'54"S, | | 2°28'30"S, | | 2°28'38"S | |
| Longitude | 29°23'20"E | | 29°20'33"E | | 29°14'36"E | | 29°12'40"E | | 29°6'53"E | |
| Elevation (m a.s.l.) | 2493 | ± 27 | 2505 | ± 31 | 2400 | ± 18 | 2415 | ± 67 | 1952 | ± 13 |
| Plot air temperatures, under canopy: | | | | | | | | | | |
| Mean annual air temperature (°C) | 13.9 | ± 0.1 | 13.7 | ± 0 | 14.2 | ± 0.5 | 14.4 | ± 0.7 | 15.6 | ± 0.2 |
| Lowest monthly mean daily minimum T (° | 11.3 | ± 0.2 | 9.9 | ± 1 | 10.1 | ± 2 | 11.7 | ± 1.3 | 10.9 | ± 0.3 |
| Highest monthly mean daily maximum T ( | 17.7 | ± 0.1 | 18.6 | ± 0 | 19.4 | ± 0.6 | 18.3 | ± 0.2 | 21.2 | ± 0.9 |
| Meterology station data, open areas: | | | | | | | | | | |
| Mean annual air temperature (°C) | 14.1 | ± 0.06 | | | 14.4 | ± 0.05 | 14.6 | ± 0.03 | 16.1 | ± 0.01 |
| Annual precipitation (mm yr$^{-1}$) | 1657 | ± 163 | | | 1755 | ± 300 | 1860 | ± 116 | 3016 | ± 63 |





**Table 2.** Biomass properties and species composition (means ± SD and range) of all plots (1-15) classified as early (ES; n = 5) or late (LS; n = 5) successional.  The data are based on results from the first census (2012) of all stems ≥ 5 cm within the plots, expect for relative growth rate, recruitment and mortality (eq. 6, 10 and 11, respectively) that were based on both census I and II data. The successional indices (based on number of stems, #, and basal area, BA) and were calculated using eq. 9. The listed species are used to construct the successional index and selected based on their contribution to total stand BA. Diff represents the mean % differences of LS in relation to ES plots and P-values are the results of a t-test of the difference between ES and LS.

| | All plots (1-15) | | | Plots of  different successional stages | | | | | |
| --- | --- | --- | --- | --- | --- | --- | --- | --- | --- |
| | | | | ES (n=5) | | LS (n=5) | | | |
| Plot properties | Mean | SD | Range | Mean | SD | Mean | SD | Diff (%) | $P$-value |
| No of stems, $D$ >5 cm (ha$^{-1}$) | 752 | ± 398 | 350 - 1844 | 645 | ± 205 | 868 | ± 563 | 35 | 0.43 |
| No of stems, $D$ >10 cm (ha$^{-1}$) | 453 | ± 218 | 220 - 958 | 421 | ± 134 | 478 | ± 277 | 14 | 0.69 |
| Mean cross sect. area, $D$ >5 cm (m$^{-2}$) | 0.047 | ± 0.025 | 0.023 - 0.109 | 0.038 | ± 0.014 | 0.050 | ± 0.034 | 33 | 0.47 |
| Mean cross sect. area, $D$ >10 cm (m$^{-2}$) | 0.073 | ± 0.036 | 0.036 - 0.158 | 0.054 | ± 0.018 | 0.082 | ± 0.046 | 50 | 0.25 |
| Basal area (BA, m$^2$ ha$^{-1}$) | 30.0 | ± 11.2 | 17.6 - 61.0 | 22.7 | ± 3.4 | 36.2 | ± 17.1 | 59 | 0.12 |
| Mean height, $D$ >5 cm (m) | 14.4 | ± 1.5 | 12.6 - 18.5 | 14.1 | ± 1.2 | 14.6 | ± 2.4 | 4 | 0.64 |
| Mean height, $D$ >10 cm (m) | 18.3 | ± 1.7 | 16.5 - 22.8 | 17.5 | ± 0.5 | 19.0 | ± 2.7 | 9 | 0.25 |
| Mean $H$ of 100 highest trees ha$^{-1}$ (m) | 24.6 | ± 3.1 | 21.7 - 33.0 | 22.2 | ± 0.4 | 26.9 | ± 4.2 | 21 | **0.040** |
| Number of big trees, $D$ >40 cm (ha$^{-1}$) | 54 | ± 36 | 26 - 172 | 39 | ± 14 | 76 | ± 56 | 94 | 0.19 |
| No of species | 22.6 | ± 10.8 | 11.0 - 49.0 | 17.0 | ± 3.2 | 29.2 | ± 14.0 | 72 | 0.093 |
| Successional index$_{\#}$ | 0.16 | ± 0.15 | 0.01 - 0.52 | 0.03 | ± 0.03 | 0.33 | ± 0.12 | 1159 | **<0.001** |
| Successional index$_{BA}$ | 0.34 | ± 0.28 | 0.002 - 0.89 | 0.03 | ± 0.02 | 0.65 | ± 0.14 | 2190 | **<0.001** |
| Most abundent species (% of BA) | | | | | | | | | |
| *Harungana montana* (ES) | 2.9 | ± 6 | 1 - 22 | 6.7 | ± 10.0 | 0.8 | ± 0.9 | | 0.23 |
| *Macaranga kilimandscharica* (ES) | 29.0 | ± 27 | 0 - 73 | 59.4 | ± 14.8 | 3.6 | ± 5.7 | | **<0.001** |
| *Polyscias fulva* (ES) | 3.7 | ± 6 | 0 - 16 | 7.9 | ± 7.2 | 0.0 | ± 0.1 | | **0.040** |
| *Carapa grandiflora* (LS) | 2.5 | ± 3 | 0 - 10 | 0.2 | ± 0.5 | 3.2 | ± 3.8 | | 0.13 |
| *Cleistanthus polystachyus* (LS) | 2.7 | ± 7 | 3 - 26 | 0.0 | ± 0.0 | 7.4 | ± 11.4 | | 0.18 |
| *Faurea saligna* (LS) | 3.2 | ± 10 | 11 - 36 | 0.0 | ± 0.0 | 9.5 | ± 15.6 | | 0.21 |
| *Ficalhoa laurifolia* (LS) | 1.4 | ± 4 | 0 - 13 | 0.1 | ± 0.2 | 4.0 | ± 5.6 | | 0.16 |
| *Ocotea kenyensis* (LS) | 2.9 | ± 3 | 5 - 23 | 4.3 | ± 4.7 | 0.0 | ± 0.0 | | 0.07 |
| *Ocotea usambarensis* (LS) | 3.0 | ± 7 | 0 - 9 | 2.7 | ± 3.7 | 4.8 | ± 3.2 | | 0.36 |
| *Syzygium guineense* (LS) | 25.3 | ± 22 | 0 - 65 | 2.2 | ± 2.3 | 38.9 | ± 19.0 | | **0.003** |
| Sum of 10 species (% of BA) | 76.6 | ± 15 | 42 - 97 | 83.6 | ± 12.9 | 72.2 | ± 13.4 | -14 | 0.21 |





**Table 3.** Wood density (basal area weighted average, ρBA), above ground biomass (AGB, stems D ≥ 5 cm), relative growth rates, recruitment rate and mortality rate of all trees with a D > 5 cm, given as means ± SD and range of all plots (1-15) classified as early (ES; n = 5) or late (LS; n = 5) successional. The AGB estimates are based on species specific or generic $H$ vs $D$ relationship and ρ.

| Plot properties | All plots (1-15) | | | Plots of different successional stages | | | | | |
| | Mean | SD | Range | ES (n=5) | | LS (n=5) | | Diff (%) | *P*-value |
| | | | | Mean | SD | Mean | SD | | |
| Wood density, specific ($\rho_{BA}$, g cm$^{-1}$) | 0.56 | ± 0.06 | 0.47 - 0.66 | 0.48 | ± 0.02 | 0.62 | ± 0.02 | 29 | **<0.001** |
| AGB, specific $H$ and ρ (Mg ha$^{-1}$) | 274 | ± 165 | 142 - 793 | 156 | ± 15 | 387 | ± 244 | 147 | **0.022** |
| AGB, generic $H$, specific ρ (Mg ha$^{-1}$) | 275 | ± 151 | 148 - 743 | 164 | ± 17 | 374 | ± 222 | 128 | **0.023** |
| AGB, specific $H$, generic ρ (Mg ha$^{-1}$) | 279 | ± 136 | 156 - 699 | 189 | ± 24 | 357 | ± 209 | 89 | 0.080 |
| AGB, generic $H$ and ρ (Mg ha$^{-1}$) | 275 | ± 125 | 161 - 650 | 191 | ± 30 | 342 | ± 189 | 79 | 0.083 |
| Relative growth rate (%, yr$^{-1}$) | 6.2 | ± 2.9 | 2.1 - 13 | 8.4 | ± 2.8 | 4.1 | ± 1.6 | -52 | **0.017** |
| Recruitment rate (%, yr$^{-1}$) | 3.8 | ± 3.4 | 0.4 - 14.1 | 6.3 | ± 4.4 | 1.4 | ± 1.0 | -77 | **0.007** |
| Mortality rate (%, yr$^{-1}$) | 1.4 | ± 0.5 | 0.5 - 2.7 | 1.1 | ± 0.4 | 1.4 | ± 0.4 | 26 | 0.26 |



**Table 4.** Carbon stocks of different ecosystem compartments (means ± SD and plot range) for all plots (1-15) classified as early (ES, n = 5) and late (LS, n = 5) successional. Diff and *P*-value represents the mean differences and the results of a t-test of the difference between ES and LS, respectively. For calculating carbon stock (C stock) in the different compartments we used measured values for each fraction of litter (37.3 - 51.3 % C), organic soil (26.0 - 49.6 % C), mineral soil (1.3 – 9.9 % C) and literature values for wood (47.4 % C, Martin and Thomas 2011) and for others we assumed 50 % C. AGB is above ground biomass; BGB is below ground biomass; $C_{Stock}$ values are in Mg C ha$^{-1}$.

| Compartment | All plots (1-15) | | | Plots of different successional stages | | | | | |
| --- | --- | --- | --- | --- | --- | --- | --- | --- | --- |
| | | | | ES (n=5) | | LS (n=5) | | | |
| | Mean SD | | Range | Mean | SD | Mean | SD | Diff (%) | *P*-value |
| $Cstock_{Stem}$ | 130 ± 78 | | 68 - 376 | 74 ± 7.3 | | 183 ± 116 | | 146 | **0.023** |
| $Cstock_{Understory}$ | 2.0 ± 1.3 | | 0.2 - 5.4 | 1.7 ± 0.6 | | 1.9 ± 1.2 | | 11 | 0.98 |
| $Cstock_{AGB}$ | 132 ± 78 | | 69.4 - 377 | 76 ± 7.2 | | 185 ± 115 | | 143 | **0.021** |
| $Cstock_{Coarse\ roots}$ | 27 ± 16 | | 14.2 - 79 | 16 ± 2 | | 38 ± 24 | | 146 | **0.023** |
| $Cstock_{Fine\ roots}$ | 3.3 ± 1.7 | | 2.0 - 8.1 | 2.8 ± 0.5 | | 3.8 ± 1.4 | | 37 | 0.17 |
| $Cstock_{BGB}$ | 31 ± 17 | | 17.5 - 83 | 18 ± 1.3 | | 42 ± 25 | | 130 | **0.025** |
| $Cstock_{Litter}$ | 4.3 ± 1.4 | | 1.6 - 6.3 | 4.8 ± 1.0 | | 3.5 ± 1.6 | | -28 | 0.15 |
| $Cstock_{Organic\ soil}$ | 31 ± 13 | | 9 - 51 | 26 ± 7 | | 36 ± 15 | | 35 | 0.34 |
| $Cstock_{Mineral\ soil}$ | 157 ± 37 | | 86 - 196 | 173 ± 13 | | 139 ± 45 | | -20 | 0.13 |
| $Cstock_{Soil\ tot}$ | 192 ± 45 | | 97 - 252 | 204 ± 13 | | 178 ± 56 | | -13 | 0.27 |
| $Cstock_{Total}$ | 353 ± 99 | | 232 - 662 | 299 ± 21 | | 402 ± 147 | | 35 | 0.11 |
| AGB fraction of $Cstock_{Total}$ (%) | 36 ± 12 | | 23 - 57 | 25 ± 0.8 | | 44 ± 14 | | 73 | **0.019** |
| AGB+BGB fraction of $Cstock_{Total}$ (%) | 44 ± 14 | | 28 - 68 | 32 ± 1.0 | | 54 ± 17 | | 70 | **0.019** |

[a]Understory data is from Ndayisabye (2014).





**Table 5.** Net primary production (NPP, eq. 5) of different forest compartments (means ± SD and range) for all plots (1-15) classified as early (ES, n = 5) and late (LS, n = 5) successional. Diff and *P*-value represents the mean differences and the results of t-test of the difference between ES and LS, respectively. For C concentrations of different compartments, see Table 4. NPP values are in Mg C ha$^{-1}$ yr$^{-1}$. OL, organic soil layer; ML, mineral soil layer.

| Compartment | All plots (1-15) | | | Plots of different successional stages | | | | |
| | | | | ES (n=5) | | LS (n=5) | | | |
| | Mean SD | | Range | Mean SD | | Mean SD | | Diff (%) | *P*-value |
|---|---|---|---|---|---|---|---|---|---|
| $NPP_{Stem}$ | 2.8 ± 1.0 | | 1.6 - 4.5 | 3.0 ± 1.2 | | 2.7 ± 1.1 | | -12 | 0.61 |
| $NPP_{CoarseRoots}$ | 0.9 ± 0.4 | | 0.5 - 1.5 | 1.0 ± 0.4 | | 0.9 ± 0.5 | | -4 | 0.89 |
| $NPP_{Wood}$ (AG & BG) | 3.7 ± 1.3 | | 2.0 - 6.0 | 4.0 ± 1.6 | | 3.6 ± 1.5 | | -10 | 0.68 |
| $NPP_{FineRoot}$ (OL) | 1.2 ± 0.4 | | 0.8 - 1.9 | 1.2 ± 0.2 | | 1.1 ± 0.4 | | -10 | 0.51 |
| $NPP_{FineRoot}$ (ML) | 0.8 ± 0.3 | | 0.2 - 1.2 | 0.8 ± 0.2 | | 0.7 ± 0.4 | | -19 | 0.41 |
| $NPP_{FineRoots}$ | 2.0 ± 0.6 | | 1.2 - 2.9 | 2.0 ± 0.4 | | 1.7 ± 0.7 | | -14 | 0.43 |
| $NPP_{Leaves}$ | 2.4 ± 0.6 | | 1.5 - 3.4 | 2.3 ± 0.4 | | 2.4 ± 0.7 | | 5 | 0.77 |
| $NPP_{Reproductive}$ | 0.5 ± 0.4 | | 0.2 - 1.6 | 0.3 ± 0.1 | | 0.6 ± 0.5 | | 63 | 0.38 |
| $NPP_{Twigs}$ | 0.5 ± 0.2 | | 0.2 - 1.0 | 0.4 ± 0.1 | | 0.6 ± 0.3 | | 39 | 0.25 |
| $NPP_{Epiphytes}$ | 0.2 ± 0.3 | | 0.01 - 1.0 | 0.2 ± 0.3 | | 0.3 ± 0.4 | | 73 | 0.58 |
| $NPP_{Other}$ | 0.04 ± 0.04 | | 0.02 - 0.2 | 0.1 ± 0.1 | | 0.03 ± 0.01 | | -52 | 0.35 |
| $NPP_{Canopy}$ | 3.7 ± 0.9 | | 2.2 - 5.6 | 3.3 ± 0.4 | | 3.9 ± 1.3 | | 18 | 0.35 |
| $NPP_{Tot}$ | 9.4 ± 1.5 | | 6.7 - 12.1 | 9.3 ± 1.7 | | 9.2 ± 2.1 | | -1 | 0.93 |
| $NPP_{Wood}/NPP_{Tot}$ (%) | 39 ± 10 | | 23 - 51 | 41.9 ± 10.9 | | 38.2 ± 9.9 | | -9 | 0.59 |
| $NPP_{FineRoots}/NPP_{Tot}$ (%) | 21 ± 6 | | 11 - 32 | 22.3 ± 5.9 | | 19.4 ± 6.7 | | -13 | 0.48 |
| $NPP_{Canopy}/NPP_{Tot}$ (%) | 40 ± 102 | | 29 - 59 | 35.8 ± 5.7 | | 42.4 ± 12.7 | | 18 | 0.32 |





**Table 6.** Above ground biomass (AGB) and forest structure (including trees with $D > 10$ cm) of old-growth tropical lowland (< 1000 m a.s.l) and montane forest of different tropical regions. The TMF sites were selected to be within an altitude range of 1600 to 2800 m a.s.l. and an annual mean temperature range of 11 to 18 °C. The mean, min and max values are based on the mean from sites. Abbreviations: SE, South-east; C & E, Central and East; C, Central; MAT, mean annual temperature, MAP, mean annual precipitation, $D$, breast height diameter; BA, basal area; ρ, wood density.

| | Tropical Lowland Forest | | | | | | | | | Tropical Montane Forest | | | | | | | | | | |
| | SE Asia[a] | | | C & E | | | C Africa[c] | | | SE Asia[d] | | | C & S America[e] | | | C & E Africa[f] | | | This study[g] |
| | Mean | Min | Max | Mean | Min | Max | Mean | Min | Max | Mean | Min | Max | Mean | Min | Max | Mean | Min | Max | Mean |
|---|---|---|---|---|---|---|---|---|---|---|---|---|---|---|---|---|---|---|---|
| Elevation (m a.s.l.) | 249 | 9 | 991 | 122 | 41 | 197 | 456 | 35 | 874 | 2205 | 1560 | 2825 | 2208 | 1750 | 2825 | 2347 | 2230 | 2464 | 2230 |
| MAT (°C) | 26 | 22 | 27 | 26 | 25 | 27 | 25 | 22 | 27 | 15 | 12 | 18 | 14 | 11 | 18 | 14 | 12 | 15 | 15 |
| MAP (mm) | 3127 | 2052 | 4441 | 2421 | 2009 | 2856 | 1853 | 1530 | 2837 | 2468 | 1891 | 3985 | 2976 | 1487 | 5000 | 2281 | 2240 | 2322 | 2322 |
| AGB (Stems, $D \geq 10$ cm) | 456 | 196 | 779 | 341 | 251 | 387 | 431 | 147 | 749 | 248 | 119 | 307 | 224 | 78 | 408 | 327 | 275 | 380 | 380 |
| BA (m ha$^{-1}$) | 37 | 22 | 49 | 29 | 23 | 34 | 32 | 14 | 47 | 41 | 34 | 53 | 36 | 27 | 51 | 42 | 35 | 49 | 35 |
| ρ (g cm$^{-3}$) | 0.60 | 0.56 | 0.64 | 0.68 | 0.65 | 0.72 | 0.64 | 0.45 | 0.84 | 0.58 | 0.56 | 0.61 | 0.54 | 0.52 | 0.56 | 0.62 | | | 0.62 |
| Stem density (ha$^{-1}$) | 584 | 326 | 1337 | 597 | - | | 426 | 181 | 650 | 1467 | 697 | 2943 | 1343 | 477 | 2753 | 428 | 378 | 478 | 478 |
| No of sites/plots/total area (ha) | 56/79/235 | | | 4/17/29 | | | 51/193/253 | | | 4/19/3 | | | 8/52/12 | | | 2/10/4 | | | 1/5/2.5 |

[a]Slik et al. (2010) - (Borneo – Brunei; Malaysia; Indonesia).

[b]Baker et al. (2004); Quesada et al. (2010); www.ctfs.si.edu/group/Ecosystems+and+Climate/Data+Resources - (Brazil).

[c]Lewis et al. (2013) - (Cameroon; Central African Republic; Democratic Republic of Congo; Gabon; Nigeria; Republic of the Congo).

[d]Aiba et al. (2005); Clumsee et al. (2010); Dossa et al. (2013); Edwards and Grubb (1977); Kitayama and Aiba (2002); Sawada et al. (2016) - (Malaysia and Indonesia; Papua New Guinea). ρ and stem density only from 3 sites.

[e]Alvarez-Alteaga et al. (2013); Delaney et al. (1997); Delaney et al. (1998); Girardin et al. (2010); Girardin et al. (2014); Grimm and Fassbender (1981); Lieberman et al. (1996); Moser et al. (2011); Leuschner et al. (2007); Spracklen et al. (2005); Unger et al. (2012) - (Costa Rica; Ecuador; Mexico; Peru; Venezuela). BA, ρ and stem density only from 3 sites.

[f]This study; Ensslin et al.(2015); Rutten et al. (2015b); Hemp (2006) - (Rwanda; Tanzania). ρ only from this study.

[g]Only LS plots, stems with $D \geq 10$ cm.





**Figure 1.** Seasonal variation of monthly mean air temperature (a) and monthly precipitation (b) at four meteorology stations located across a 32 km east-west transect in Nyungwe tropical montane forest. Roman numbers refers to locations presented in Table 1. The data are based on half-hourly measurements from July 2013 to June 2015.

5 **Figure 2.** Distribution of the mean stem numbers (a) and stem biomass (b) per unit area in relation to $D$ classes for three groups of plots (n= 5 for all groups) belonging to different successional stages. The error bars are standard deviation (SD).

**Figure 3.** Height ($H$) vs stem diameter at breast height $D$ relationship for 930 trees representing the 25 most abundant species of all plots and fitted to equation 2. The relationship for all measurements (red line) is compared to the measurements of 1982 trees in African tropical forests (Lewis et al 2009, grey dashed line) mainly at an altitude below 1000 m.a.s.l (a) as 10 well as to species-specific functions for the 10 most abundant species, including (b) three early successional species (ES) and (c) seven late successional species (LS). All species specific data are presented in Table S3. The mean simulated height (m) at a $D = 10$ cm was $13.4 \pm 3.3$ in ES and $12.0 \pm 1.4$ in LS ($P = 0.35$); $D = 40$ cm was $23.3 \pm 1.5$ in ES and $25.9 \pm 1.5$ in LS ($P = 0.033$); $D = 80$ cm was $26.4 \pm 1.7$ in ES and $33.4 \pm 3.9$ in LS ($P = 0.020$).

**Figure 4.** Annual stem volume increment (a), mass increment (b) and relative growth rate (c) for *M. kilimandscharica* (n = 15 112) and *S. guineense* (n = 119) distributed over all available $D$ classes and all plots. Stem production estimates are based on 9 consecutive recordings over three years of $D$ with fixed dendrometer bands. The volume increment, mass increment and relative growth rate (RGR) of stems average over the six lowest $D$ classes (10 to 70 cm) were 53 % ($P = 0.012$), 6 % ($P = 0.62$) and 35 % ($P = 0.027$) larger for *M. kilimandscharica* compared to *S. guineense*. A mean weighted based on $D$ classes was used in the comparison.

20 **Figure 5.** Biomass, relative growth rates (RGR) and net primary production (NPP) of stems including branches of each plot in relation to its successional index, based on basal area (BA) of most abundant early and late successional tree species (eq. 10). Biomass, RGR and NPP are based on C units and calculated from eq. 3, 6, 5, respectively, based on measurements of $D_{BH}$, height, wood density and a stem tissue C concentration of 47.4 %.

**Figure 6.** Biomass of stems including branches in relation to understory index (a); number of big trees with a $D > 40$ cm (b) 25 and mean relative growth rates (RGR) in relation basal area (BA). Understory index is calculated from eq. 8 and biomass and RGR are expressed in C units and calculated from eq. 3, 6 respectively, based on measurements of $D$, $H$ and $\rho$ and a stem tissue C concentration of 47.4 %. The correlation of stem biomass *vs*. number of big trees (b) is also significant when the extreme value is omitted (P = 0.004).





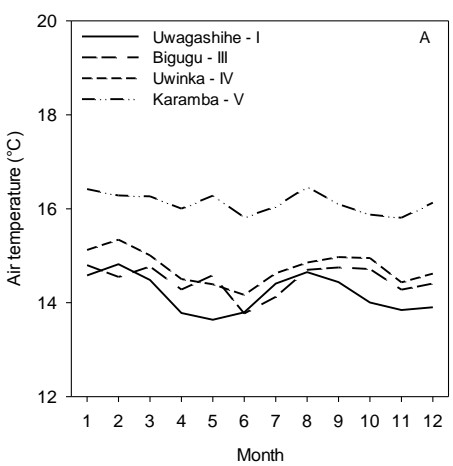
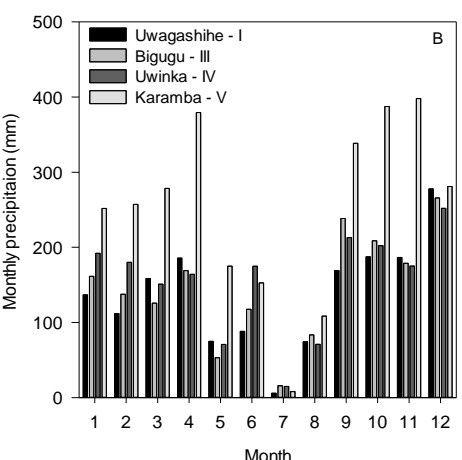

5    **Figure 1.**





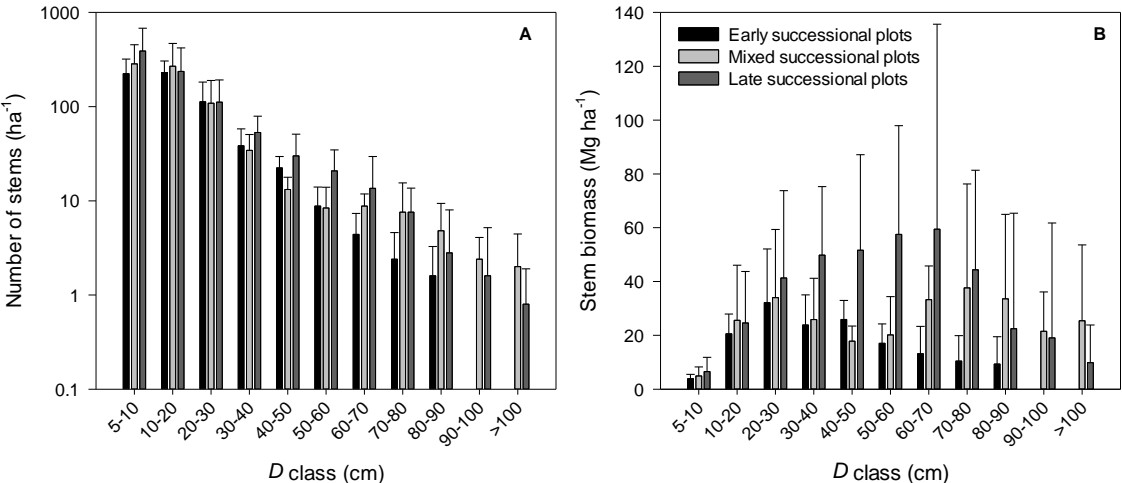

**Figure 2.**



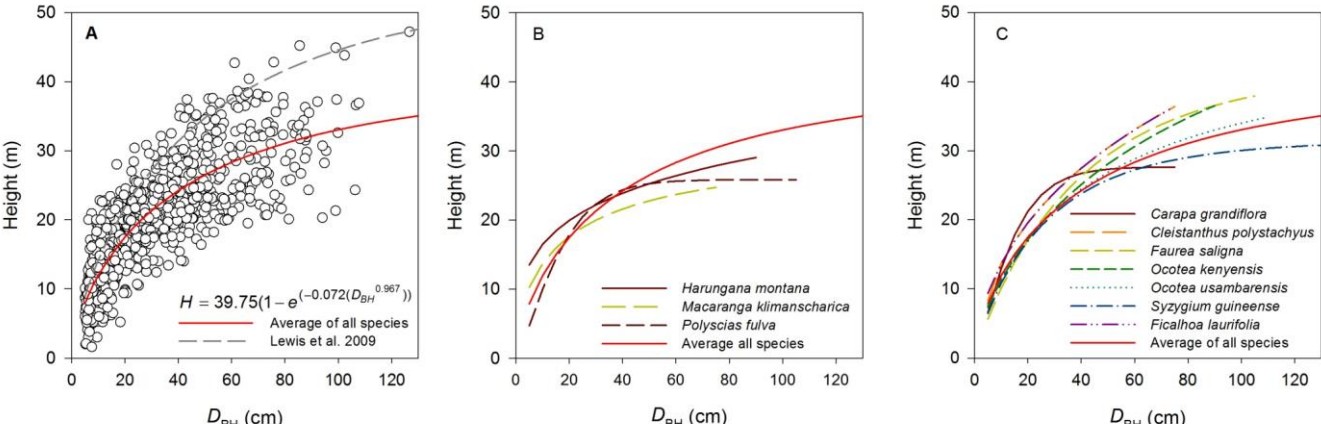

**Figure 3.**





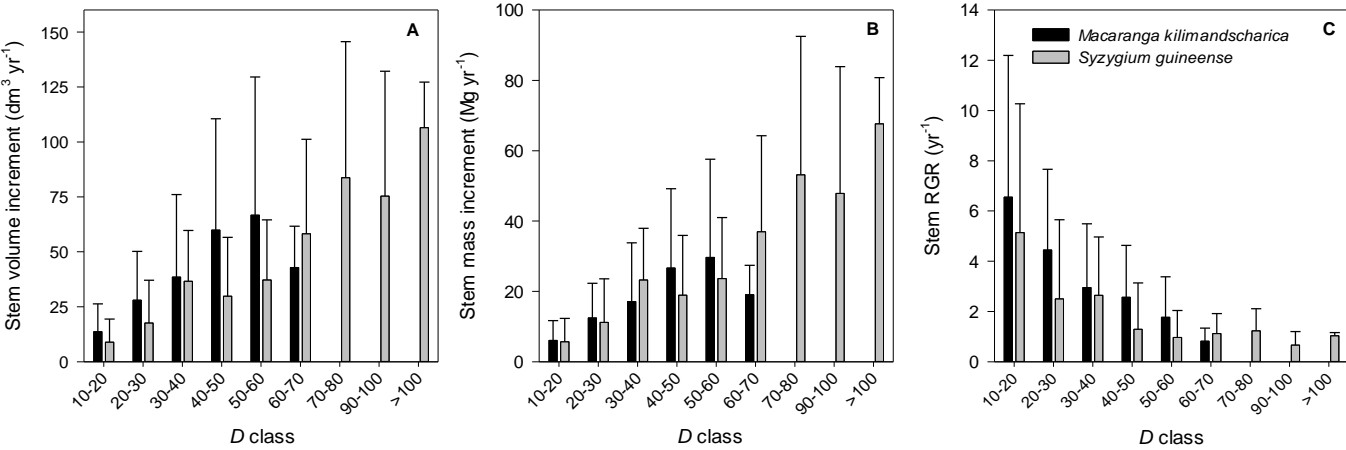

Figure 4.

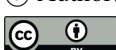



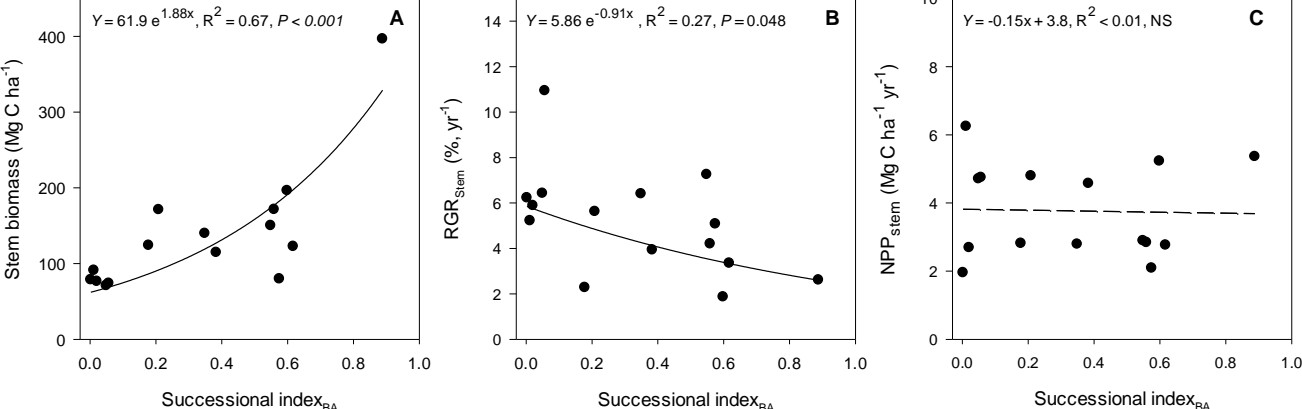

**Figure 5.**





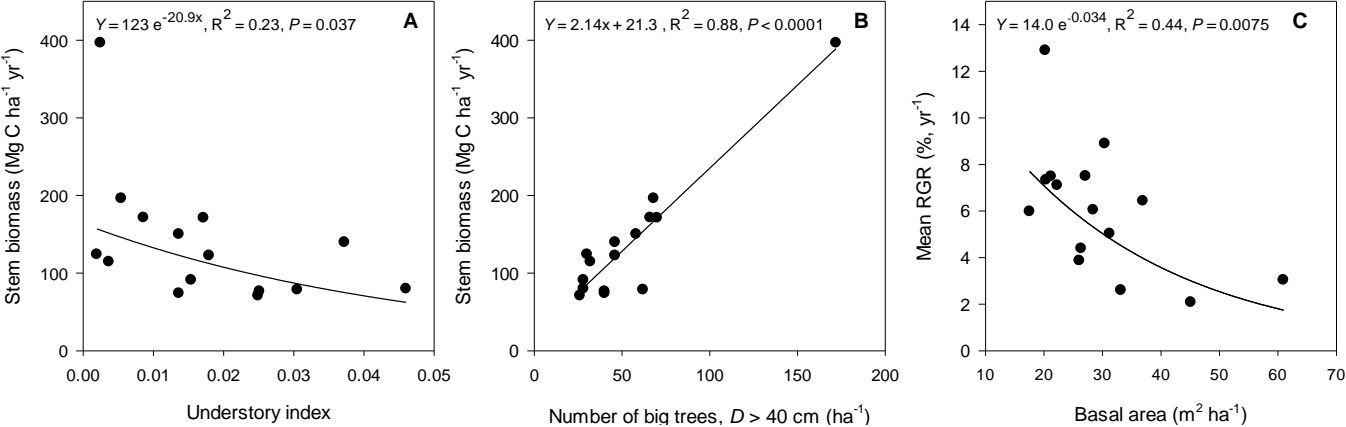

Figure 6.