# Peer review of "Carbon stocks and dynamics at different successional stages in an Afromontane tropical forest"

_Biogeosciences, 2016_

## Referee Comment (RC1) · G. Moser (Referee) · 6 Oct 2016

This manuscript fills an existing gap about the carbon pools of African montane forests of different stages from early to late succession. The authors have applied up to date methods to determine the above- and belowground biomass and productivity and soil carbon pools and found that the late successional forests investigated have higher carbon pools than montane forests in S-America and SE-Asia. The manuscript title clearly reflects the content, the abstract gives a nice overview and summary. The paper is well structured and presented, written in fluent and precise English. Formulas and abbreviations are correctly given and used.

The methods are in general clearly described with only view details, that should be

clarified: In the description of the study area and the plots I was missing information about the slope inclination and topographic position of the plots. Only the planimetric area is given, but especially the topographic position (ridge, valey bottom) would be an interesting information, as it is known to have an important influence on forest structure. Later in the methods it would be good to clarify that all area-based results (biomass, NPP, C-pools) are related to the planimetric area and not to the inclinated area.

In section 2.3 Meteorological data the authors list ... air humidity, solar radiation, humidity ... Please clarify what is meant by "humidity", maybe soil moisture? A list of meteorological parameters that were measured are not presented like solar radiation, soil temperature and moisture. So, I would like to ask the authors to present these data. The authors used ingrowth cores to quantify the fine root production. I was wondering if they determined the lack time between installation of the ingrowth core and the time when roots started after the disturbance to grow inside the root free soil cores. Please clarify if you determined the lack time and subtracted these periods for annual fine root productivity, or if you didn't.

The results are well and traceable presented and support the conclusions. I In Tab 2 – 5 I would like to ask the authors to present also the results of the intermediate successional plots MS. To me that information would be of more interest than the total mean and SD of all 15 plots. In line 2 of Table 6 "America b" is missing after C&E. And please replace Clumsee et al. by Culmsee et al. On page 12 lines 14-15 it remains unclear if the given recruitment rates refer to the above mentioned species or not.

Putting their own work into the context of published studies the authors clearly indicate the origin of the data. Only a view studies are missing in their review on forest structure and above and belowground biomass. Here the authors should include additional results from SE-Asia from Hertel et al. 2010 Forest Ecology and Management and from Kessler et al 2012 PloSOne. On page 14 line 29 it should say " M. kilimandscharica"

The supplementary material presents important detail information and is well presented. In Table S3 please correct in the first line "species abundance". In Fig S1 please give also R2adj when non-linear correlations were applied. This manuscript will be an important contribution to Biogeosciences as the results fill still existing gaps about the C pools and dynamics of Afromontane forests of different successional stages. It is nice to read, well presented and comes to interesting conclusions based on a great dataset.

---

## Referee Comment (RC2) · Anonymous Referee #2 · 4 Nov 2016

This study reports measurements of biomass pools in early and late successional tropical montane forests in Rwanda, Africa. The data in this manuscript is an important addition to the very sparse data on carbon stocks in African montane forests. The manuscript reports biomass pools and relative growth rates and compares early to late successional forests. This study adds to our knowledge of biomass storage during succession in tropical montane forests. The methods are well described and the analysis is appropriate. The manuscript is clearly and concisely written. I recommend publication and I only have very minor comments as detailed below.

Minor comments

Page 4. A map showing the study area might be useful for the reader.

[Figure]

Page 4. Do you have information on the slope of the plots? If so, it would be useful to include.

Page 4. Do you have an idea of the major cause of disturbance at the studied plots? It would be useful to comment on this. Is the disturbance likely natural or anthropogenic?

Table 1. Minor formatting issues in the "Properties" column.

---

## Author Comment (AC1) · 1 Dec 2016

Thank you for your comments on our Biogeosciences discussion paper. Below you have our response to each comment. We have also prepared a revised manuscript according to the comments and response (see Supplement), ready to be submitted

Comments from Referee 1

This manuscript fills an existing gap about the carbon pools of African montane forests of different stages from early to late succession. The authors have applied up to date methods to determine the above- and belowground biomass and productivity and soil carbon pools and found that the late successional forests investigated have higher

carbon pools than montane forests in S-America and SE-Asia. The manuscript title clearly reflects the content, the abstract gives a nice overview and summary. The paper is well structured and presented, written in fluent and precise English. Formulas and abbreviations are correctly given and used. The methods are in general clearly described with only view details, that should be clarified: In the description of the study area and the plots I was missing information about:

RESPONSE: Thanks for kind comments. Below we answer all specific questions and comments.

1. The slope inclination and topographic position of the plots. Only the planimetric area is given, but especially the topographic position (ridge, valey bottom) would be an interesting information, as it is known to have an important influence on forest structure.

RESPONSE: We have included information on slope and topographic position in Table S1.

2. Later in the methods it would be good to clarify that all area-based results (biomass, NPP, C-pools) are related to the planimetric area and not to the inclinated area.

RESPONSE: We have included a sentence at the end of section 2.2 "All forest area based information is related to the planimetric area."

3. In section 2.3 Meteorological data the authors list . . . air humidity, solar radiation, humidity . . . Please clarify what is meant by "humidity", maybe soil moisture?

RESPONSE: Unfortunately, humidity was repeated twice. The second "humidity" was a typing error and has been deleted.

4. A list of meteorological parameters that were measured are not presented like solar radiation, soil temperature and moisture. So, I would like to ask the authors to present these data.

RESPONSE: We added two sentences to report soil temperature and solar radiation

in section 3.1. Soil water content was already reported in the same section. "The daily mean soil temperatures were similar to the air temperatures, but with lower mean diurnal amplitudes (1.9 °C in the soil compared to 6.1°C in air)." "The daily mean photosynthetic photon flux density at the four meteorology stations was $289 \pm 10 \ \mu$mol m-2 s-1, with slightly elevated levels during the dry period."

5. The authors used ingrowth cores to quantify the fine root production. I was wondering if they determined the lack time between installation of the ingrowth core and the time when roots started after the disturbance to grow inside the root free soil cores. Please clarify if you determined the lack time and subtracted these periods for annual fine root productivity, or if you didn't.

RESPONSE: We guess that you mean lag-time here, however we did not include any lag-time which might have underestimated the growth. However, since we used relatively long intervals (6 month) between the harvests, we assume that any lag effect would be of smaller importance in this study compared to the commonly used, 3-4 month intervals. To clarify this we added the following information in section 2.7: "The annual sum of production was calculated in proportion to the time between the harvests (without assumption of any lag-time)"

6. The results are well and traceable presented and support the conclusions. I In Tab 2 – 5 I would like to ask the authors to present also the results of the intermediate successional plots MS. To me that information would be of more interest than the total mean and SD of all 15 plots.

RESPONSE: We believe that the overall message is clearer by mostly presenting data from the two distinctly separated ES and LS groups. However, to show that there is a successional transition between ES and LS plots we suggest extending Table S1 with calculations of the mean for all three groups statistically analysed with a one-way ANOVA to show how MS relates to ES and LS for some of the most important biomass parameters. We also added references to Table S1 regarding MS values in section 3.3,

3.4 and 3.5.

7. In line 2 of Table 6 "America b" is missing after C&E

RESPONSE: Thanks for pointing out the missing note. It has been added, however it is C & E Amazonia.

8. And please replace Clumsee et al. by Culmsee et al.

RESPONSE: Thanks for noting the spelling error. It has been corrected.

9. On page 12 lines 14-15 it remains unclear if the given recruitment rates refer to the above mentioned species or not.

RESPONSE: Yes, also recruitment rate refer to the two species M. kilimandscharica and S. guineense. We have revised the sentence to become more clear, and deleted a typing error, as follows: "However, recruitment rate (3.0/1.4%, P = 0.17) did not differ significantly differ between these two species on when analysing plots where they co-occurred.

10. Putting their own work into the context of published studies the authors clearly indicate the origin of the data. Only a view studies are missing in their review on forest structure and above and belowground biomass. Here the authors should include additional results from SE-Asia from Hertel et al. 2010 Forest Ecology and Management and from Kessler et al 2012 PloSOne.

RESPONSE: The two suggested papers are definitely two important studies of stocks and production of biomass in tropical forest. However, the reason why we did not include these in our overview is that they are conducted at around 1000 m a.s.l., while we focused our comparisons with other tropical montane forests on studies conducted at an elevation range of 1600-3000 m, and our lowland overview included only studies conducted < 1000 m a.s.l.

11.On page 14 line 29 it should say " M. kilimandscharica"

RESPONSE: Thank you for observing this spelling error. It has been corrected.

12.The supplementary material presents important detail information and is well presented. In Table S3 please correct in the first line "species abundance".

RESPONSE: Thank you for observing this spelling error. It has been corrected.

13.In Fig S1 please give also R2adj when non-linear correlations were applied.

RESPONSE: We have included adjusted R2 values in the legends of all figures with non-linear functions.

14.This manuscript will be an important contribution to Biogeosciences as the results fill still existing gaps about the C pools and dynamics of Afromontane forests of different successional stages. It is nice to read, well presented and comes to interesting conclusions based on a great dataset.

RESPONSE: Thanks!

Please also note the supplement to this comment:
http://www.biogeosciences-discuss.net/bg-2016-353/bg-2016-353-AC1-supplement.pdf

**Supplement:**

[revised manuscript text omitted]

**Supplementary information** (Nyirambangutse et al.)**:**

**Table S1.** Characteristics  of all plots. SG, successional group including early (ES), mixed (MS) and late (LS); TG , topography including ridge (R), upper slope (US), slope (S) and valley (V); Slope, average slope of all subplot sides; Breast height diameter ($D$) are measured or calculated from diameter measurements above any stem irregularities (eq. 1). Height ($H$) was measured on 930 trees to obtain coefficients to calculate $H$ (presented here) from $D$ of all trees (eq. 2). No of stems, basal area (BA), mean $D$, mean height ($H$), stem biomass and production (prod) includes branches and were based on data from all trees > 5 cm $D_{BH}$. Big trees are > 40 cm $D$. Stem biomass was calculated from eq. 3 based on four different combinations of species specific (Spec.) and generic (Gen.) $H$ vs $D$ relationships and wood densities ($\rho$). RGR, relative growth rate, mortality ($\mu$) and recruitment ($\lambda$) rate were calculated from eq. 6, 10 and 11 respectively. $SI_{BA}$ and $SI_{\#}$, successional index based on BA or no of stems (#) was calculated from eq. 9.

[revised manuscript text omitted]

---

## Author Comment (AC2) · 1 Dec 2016

Thank you for your comments on our Biogeosciences discussion paper. Below you have our response to each comment. We have also prepared a revised manuscript according to the comments and response, ready to be submitted.

Comments from Referee 2:

This study reports measurements of biomass pools in early and late successional tropical montane forests in Rwanda, Africa. The data in this manuscript is an important addition to the very sparse data on carbon stocks in African montane forests. The manuscript reports biomass pools and relative growth rates and compares early to

late successional forests. This study adds to our knowledge of biomass storage during succession in tropical montane forests. The methods are well described and the analysis is appropriate. The manuscript is clearly and concisely written. I recommend publication and I only have very minor comments as detailed below.

RESPONSE: Thank you for kind comments

Minor comments:

Page 4. A map showing the study area might be useful for the reader.

RESPONSE : We have included a map in the supplementary material

Page 4. Do you have information on the slope of the plots? If so, it would be useful to include.

RESPONSE: Yes, average slopes of all plots have now been included in Table S1.

Page 4. Do you have an idea of the major cause of disturbance at the studied plots? It would be useful to comment on this. Is the disturbance likely natural or anthropogenic? RESPONSE: In section 2.1 we write: The secondary forest areas are mainly created from human induced disturbance such as tree cutting, fire, and mining, but natural disturbances such as landslide and fallen trees are also significant. Unfortunately, the disturbance in the past was not monitored so it is difficult to describe the disturbance history in more detail.

Table 1. Minor formatting issues in the "Properties" column.

RESPONSE: We have changed the format of the first column so all text can fit.

Please also note the supplement to this comment:
http://www.biogeosciences-discuss.net/bg-2016-353/bg-2016-353-AC2-supplement.pdf

[Figure]

**Supplement:**

[revised manuscript text omitted]

**Supplementary information** (Nyirambangutse et al.)**:**

**Table S1.** Characteristics  of all plots. SG, successional group including early (ES), mixed (MS) and late (LS); TG , topography including ridge (R), upper slope (US), slope (S) and valley (V); Slope, average slope of all subplot sides; Breast height diameter ($D$) are measured or calculated from diameter measurements above any stem irregularities (eq. 1). Height ($H$) was measured on 930 trees to obtain coefficients to calculate $H$ (presented here) from $D$ of all trees (eq. 2). No of stems, basal area (BA), mean $D$, mean height ($H$), stem biomass and production (prod) includes branches and were based on data from all trees > 5 cm $D_{BH}$. Big trees are > 40 cm $D$. Stem biomass was calculated from eq. 3 based on four different combinations of species specific (Spec.) and generic (Gen.) $H$ vs $D$ relationships and wood densities ($\rho$). RGR, relative growth rate, mortality ($\mu$) and recruitment ($\lambda$) rate were calculated from eq. 6, 10 and 11 respectively. $SI_{BA}$ and $SI_{\#}$, successional index based on BA or no of stems (#) was calculated from eq. 9.

[revised manuscript text omitted]

---

## Author Response (AR1)

Dear Associate Editor!

Thank you for your comments on our Biogeosciences discussion paper and our respone to the referees. Below you have a point-by-point response to the comments by referees. We have also prepared a revised manuscript according to the comments and response, both in a version where all changes are visible and a clean copy. The authors have also reviewed the papers and we have accordingly corrected a few mistakes and improved the language a bit.

On behalf of the authors

Brigitte Nyirambangutse

| Comments from Referee 1: | Response from Authors |
|---|---|
| This manuscript fills an existing gap about the carbon pools of African montane forests of different stages from early to late succession. The authors have applied up to date methods to determine the above- and belowground biomass and productivity and soil carbon pools and found that the late successional forests investigated have higher carbon pools than montane forests in S-America and SE-Asia. The manuscript title clearly reflects the content, the abstract gives a nice overview and summary. The paper is well structured and presented, written in fluent and precise English. Formulas and abbreviations are correctly given and used. The methods are in general clearly described with only view details, that should be clarified: In the description of the study area and the plots I was missing information about: | Thanks for kind comments. Below we answer all specific questions and comments. |
| 1. The slope inclination and topographic position of the plots. Only the planimetric area is given, but especially the topographic position (ridge, valey bottom) would be an interesting information, as it is known to have an important influence on forest structure. | We have included information on slope and topographic position in Table S1. |
| 2. Later in the methods it would be good to clarify that all area-based results (biomass, NPP, C-pools) are related to the planimetric area and not to the inclinated area. | We have included a sentence at the end of section 2.2 "All forest area based information is related to the planimetric area." |

| | |
|---|---|
| 3. In section 2.3 Meteorological data the authors list . . . air humidity, solar radiation, humidity . . . Please clarify what is meant by "humidity", maybe soil moisture? | Unfortunately, humidity was repeated twice. The second "humidity" was a typing error and has been deleted. |
| 4. A list of meteorological parameters that were measured are not presented like solar radiation, soil temperature and moisture. So, I would like to ask the authors to present these data. | We added two sentences to report soil temperature and solar radiation in section 3.1. Soil water content was already reported in the same section. "The daily mean soil temperatures were similar to the air temperatures, but with lower mean diurnal amplitudes (1.9 °C in the soil compared to 6.1°C in air)." "The daily mean photosynthetic photon flux density at the four meteorology stations was 289 ± 10 µmol m$^{-2}$ s$^{-1}$, with slightly elevated levels during the dry period." |
| 5. The authors used ingrowth cores to quantify the fine root production. I was wondering if they determined the lack time between installation of the ingrowth core and the time when roots started after the disturbance to grow inside the root free soil cores. Please clarify if you determined the lack time and subtracted these periods for annual fine root productivity, or if you didn't. | We guess that you mean lag-time here, however we did not include any lag-time which might have underestimated the growth. However, since we used relatively long intervals (6 month) between the harvests, we assume that any lag effect would be of smaller importance in this study compared to the commonly used, 3-4 month intervals. To clarify this we added the following information in section 2.7: "The annual sum of production was calculated in proportion to the time between the harvests (without assumption of any lag-time)" |
| 6. The results are well and traceable presented and support the conclusions. I In Tab 2 – 5 I would like to ask the authors to present also the results of the intermediate successional plots MS. To me that information would be of more interest than the total mean and SD of all 15 plots. | We believe that the overall message is clearer by mostly presenting data from the two distinctly separated ES and LS groups. However, to show that there is a successional transition between ES and LS plots we suggest extending Table S1 with calculations of the mean for all three groups statistically analysed with a one-way ANOVA to show how MS relates to ES and LS for some of the most important biomass parameters. We also added references to Table S1 regarding MS values in section 3.3, 3.4 and 3.5. |
| 7. In line 2 of Table 6 "America b" is missing after C&E | Thanks for pointing out the missing note. It has been added, however it is C & E Amazonia. |
| 8. And please replace Clumsee et al. by Culmsee et al. | Thanks for noting the spelling error. It has been corrected. |
| 9. On page 12 lines 14-15 it remains unclear if the given recruitment rates refer to the above mentioned species or not. | Yes, also recruitment rate refer to the two species *M. kilimandscharica* and *S. guineense.* We have revised the sentence to become more clear, and deleted a typing error, as follows: "However, recruitment rate (3.0/1.4%, *P* = 0.17) did not  significantly differ between these two species on  plots where they co-occurred. |

| | |
|---|---|
| 10. Putting their own work into the context of published studies the authors clearly indicate the origin of the data. Only a view studies are missing in their review on forest structure and above and belowground biomass. Here the authors should include additional results from SE-Asia from Hertel et al. 2010 Forest Ecology and Management and from Kessler et al 2012 PloSOne. | According to the recommendations, we have cited the two suggested articles in the discussion: Page 14 row 9-11: Our values on canopy and stem NPP were similar to what was found in a lower montane forest (1050 m a.s.l.) in South-East Asia (Hertel et al. 2009), while our fine root NPP was approximately twice as high as in that study. Page 16 row 14-15: "…while much higher than in a natural submotane rainforest in Sulawesi having only 32% of the C in the soil (Kessler et al. 2012)." |
| 11. On page 14 line 29 it should say " M. kilimandscharica" | Thank you for observing this spelling error. It has been corrected. |
| 12. The supplementary material presents important detail information and is well presented. In Table S3 please correct in the first line "species abundance". | Thank you for observing this spelling error. It has been corrected. |
| 13. In Fig S1 please give also R2adj when non-linear correlations were applied. | We have included adjusted $R^2$ values in the legends of all figures with non-linear functions. |
| 14. This manuscript will be an important contribution to Biogeosciences as the results fill still existing gaps about the C pools and dynamics of Afromontane forests of different successional stages. It is nice to read, well presented and comes to interesting conclusions based on a great dataset. | Thanks! |

| Comments from Referee 2: | Response from Authors |
|---|---|
| This study reports measurements of biomass pools in early and late successional tropical montane forests in Rwanda, Africa. The data in this manuscript is an important addition to the very sparse data on carbon stocks in African montane forests. The manuscript reports biomass pools and relative growth rates and compares early to late successional forests. This study adds to our knowledge of biomass storage during succession in tropical montane forests. The methods are well described and the analysis is appropriate. The manuscript is clearly and concisely written. I recommend publication and I only have very minor comments as detailed below. | Thank you for kind comments |

| Minor comments: Page 4. A map showing the study area might be useful for the reader. | We have included a map in the supplementary material |
|---|---|
| Page 4. Do you have information on the slope of the plots? If so, it would be useful to include. | Yes, average slopes of all plots have now been included in Table S1. |
| Page 4. Do you have an idea of the major cause of disturbance at the studied plots? It would be useful to comment on this. Is the disturbance likely natural or anthropogenic? | In section 2.1 we write: The secondary forest areas are mainly created from human induced disturbance such as tree cutting, fire, and mining, but natural disturbances such as landslide and fallen trees are also significant. Unfortunately, the disturbance in the past was not monitored so it is difficult to describe the disturbance history in more detail. |
| Table 1. Minor formatting issues in the "Properties" column. | We have changed the format of the first column so all text can fit. |

[revised manuscript text omitted]

5   **Figure 3.**

[Figure]

Figure 4.

[Figure]

**Figure 5.**

[Figure]

Figure 6.